



# Occurrence of Polar Stratospheric Clouds as derived from Ground-based Zenith DOAS Observations using the Colour Index

Bianca Lauster[1,2], Steffen Dörner[1], Carl-Fredrik Enell[3], Udo Frieß[2], Myojeong Gu[1], Janis Puķīte[1], Uwe Raffalski[4], and Thomas Wagner[1]

[1]Max Planck Institute for Chemistry, Mainz, Germany
[2]University of Heidelberg, Heidelberg, Germany
[3]EISCAT Scientific Association, Kiruna, Sweden
[4]Swedish Institute of Space Physics, Kiruna, Sweden

**Correspondence:** Bianca Lauster (b.lauster@mpic.de)

**Abstract.** Polar stratospheric clouds (PSCs) are an important component of ozone chemistry in polar regions. Studying the ozone-depleting processes requires a precise description of PSCs on a long-term basis. Although satellite observations already yield high spatial coverage, continuous ground-based measurements covering long time periods can be a valuable complement. In this study, differential optical absorption spectroscopy (DOAS) instruments are used to investigate the occurrence of PSCs

based on the so-called colour index (CI), i.e. the colour of the zenith sky. Defined as the ratio between the observed intensities of scattered sunlight at two wavelengths, it provides a method to detect PSCs during twilight even in the presence of tropospheric clouds. We present data from instruments at the German research station Neumayer, Antarctica (71 °S, 8 °W) as well as Kiruna, Sweden (68 °N, 20 °E), which are in operation for more than 20 years. For a comprehensive interpretation of the measurement data, the well-established radiative transfer model McArtim is used and radiances of scattered sunlight are simulated at several

wavelengths for different solar zenith angles and various atmospheric conditions. The aim is to improve and evaluate the potential of this method. It is then used to infer the seasonal cycle and the variability of PSC occurrence throughout the time series measured in both hemispheres. A good agreement is found to satellite retrievals with deviations particularly in spring. The unexpectedly high signal observed in the DOAS data during spring time suggests the influence of volcanic aerosol. This is also indicated by enhanced aerosol extinction as seen from OMPS data but is not captured by other PSC climatologies.

The presented approach allows the detection of PSCs for various atmospheric conditions not only for individual case studies but over entire time series, which is a decisive advance compared to previous work on the PSC detection from ground-based instruments. Apart from the interannual variability, no significant trend is detected for either measurement station. The averaged PSC relative frequency amounts to about 37 % above the Neumayer station and about 18 % above Kiruna.

## 1   Introduction

The discovery of the Antarctic ozone hole in the 1980s increased the scientific interest in polar stratospheric clouds (PSCs). The cloud particles provide surfaces for heterogeneous reactions which convert halogen reservoir species into radicals and are



thus an important component of the catalytic ozone destruction in the polar stratosphere (e.g., Solomon et al., 1986; McElroy et al., 1986).

Polar stratospheric clouds appear as nacreous clouds particularly above Antarctica and near large longitudinal mountain ranges in the northern hemisphere, e.g. the mountains of Scandinavia. They typically form at altitudes of about 20 to 25 km in the winter polar stratosphere where temperatures drop below the formation threshold. However, due to nucleation barriers, temperatures below the existence temperature are necessary for the formation of PSC particles. In the course of the winter, the altitude of PSC layers systematically decreases, corresponding to the changing temperature profiles (von Savigny et al., 2005). In general, PSCs are distinguished based on their particle composition and characteristics. They are composed of supercooled

ternary solution (STS), nitric acid trihydrate (NAT) and/or ice particles, ranging from about $0.3\,\mu$m up to $\geqslant 10\,\mu$m (Tritscher et al., 2021, and references therein).

The expected climatic changes in the stratosphere might have an impact on the occurrence of PSCs. For example, variations in stratospheric water vapour affect ozone as well as stratospheric temperatures (Shindell, 2001). Thus, it is especially important to monitor PSCs on a long term basis.

Although satellites already provide measurements of PSCs since the launch of the SAM-II (Stratospheric Aerosol Measurement II) instrument on-board the Nimbus 7 satellite (McCormick et al., 1982), consistent measurement time series also from ground-based instruments are important. Since lidar stations are often influenced by tropospheric clouds, differential optical absorption spectroscopy (DOAS) measurements are a valuable complement to the other data sets.

The time series presented in this work comprise data from two stations in both hemispheres (Neumayer, Antarctica and

Kiruna, Sweden) and spanning over 20 years of almost continuous observations. The two measurement stations and the different data sets used in this study are introduced in the following section. The method is based on the so-called colour index (CI) which is defined as the ratio of radiances at two different wavelengths and was first introduced by Sarkissian et al. (1991). Especially at high solar zenith angles (SZA), PSCs can strongly influence the atmospheric radiative transfer. Thus, they can be detected even by ground-based instruments measuring zenith scattered sun light. Depending on the altitude, optical properties and the

considered wavelengths, PSCs can enhance or decrease the measured radiances compared to the non-PSC case. The method is detailed in Sect. 3. For a quantitative interpretation, measurements need to be compared to radiative transfer simulations. Here, the radiative transfer model McArtim (Deutschmann et al., 2011) is used to investigate the advantages and disadvantages of different wavelength pairs for the calculation of the CI (Sect. 3.1). A specific aspect that is investigated in this study is the effect of whether PSCs are localised (such as lee wave PSCs) or horizontally extended. The influence of additional tropospheric

cloud layers is also studied. The comparison to measurement data is given in Sect. 3.2. Based on the findings of the above mentioned studies, the method is applied to detect PSCs above both stations. Section 4 discusses the typical features of the PSC seasonal cycle and compares the results with satellite retrievals (Sect. 4.1). The main part comprises a detailed analysis of the measurement time series from the two stations (Sect. 5 and Sect. 6). Thereafter, in Sect. 6.1 the influence of volcanic aerosol on the data is discussed. Conclusions can be found in Sect. 7.



## 2 Instrumentation


In this study, ground-based DOAS instruments were used to detect PSCs. Originally, these instruments are designed to retrieve information on the abundance of trace gases such as $NO_2$, $BrO$ and $OClO$. Here, only the differential slant column densities (DSCDs) of the oxygen collision complex $O_4$ are used for filtering tropospheric cloud events. The algorithm is detailed in the Appendix A. The detection of PSC cases is based on the CI calculated for selected wavelengths of the measured spectra.

We present data from two measurement stations. One DOAS instrument is located at the German research station Neumayer on the Ekström ice shelf in the Atlantic sector of the Antarctic continent about 71 °S and 8 °W. The set-up consists of two temperature-stabilised spectrometers in the UV (320 nm to 422 nm) and visible (397 nm to 653 nm) spectral range. It is almost continuously operating since 1999 and a detailed description can be found in Frieß et al. (2001, 2005). At the end of 2002, a change in the instrument set-up led to a change in the light throughput of the spectrometer. Although set up as MAX (Multi

AXis)-DOAS, only zenith looking measurements are considered in this study.

The other instrument is a zenith-sky DOAS in Kiruna, Sweden, at about 68 °N and 20 °E. It was set-up in December 1996. The detector system was moved inside the institute building in 1999, which led to a stable air temperature and thus also more stable instrument operation. Due to technical issues, there is no usable data for the winter seasons between 2006 and 2013, after which a new detector system was installed that has lower detector noise. The temperature-stabilised spectrometer measures in

the UV spectral range from 267 nm to 397 nm. Details on the instrument are given in Otten et al. (1998), Bugarski (2003) and Gottschalk (2013).

The wavelength calibration is conducted once for each data set, i.e. for each spectrometer, using the spectrum of a mercury lamp (in case of the Neumayer DOAS) or a sun spectrum (in case of the Kiruna DOAS). For the presented application, no radiometric calibration is needed. Deviations throughout the time series are negligible. Moreover, there is no indication of

instrument degradation for either instrument, as seen from an examination of the spectra and negligible interannual variations during the summer months. Furthermore, all spectra are corrected for an instrumental offset and thermally induced dark current.

For a comprehensive interpretation of the measurement data, radiative transfer simulations are needed to investigate the effect of PSCs on the spectra for different particle properties and cloud scenarios. Here, the well-established Monte Carlo radiative transfer model McArtim developed by Deutschmann et al. (2011, version of 21 April 2014) is used to compute the

radiances at different wavelengths. The details of the different parameter settings are introduced in Sect. 3.1.

In addition, meteorological data from ECMWF (European Centre for Medium-Range Weather Forecasts) are considered in the vicinity of the measurement stations. Data sets are compiled from ERA-Interim (Dee et al., 2011) for the period before 2019 and ERA-5 (Hersbach et al., 2020) for the remaining time period because these data sets were readily available. Data are averaged in the area from 70 °S to 71 °S and 8 °W to 10 °W for the Neumayer station, as well as 67 °N to 69 °N and 19 °E to

21 °E for the Kiruna station. Data is further averaged on a half-daily basis, i.e. the 0 UTC and 6 UTC output is associated to sunrise measurements while the 12 UTC and 18 UTC output is associated to sunset measurements.

Moreover, satellite retrievals of the PSC occurrences are used for a comparison to the DOAS measurements. In this study, we focus on the PSC climatologies from CALIOP (Cloud Aerosol Lidar with Orthogonal Polarization) and MIPAS (Michel-





son Interferometer for Passive Atmospheric Sounding). CALIOP is a polarisation-sensitive lidar system on-board CALIPSO
which is in orbit since 2006. The precise measurements of cloud and aerosol profiles with a horizontal resolution below 5 km
(Thomason et al., 2007) allow the investigation of PSCs from 4 km above the reported tropopause height (Pitts et al., 2009).
MIPAS operated from 2002 to 2012 on-board Envisat and yields information on PSCs from infrared emission spectra (Spang
et al., 2016, 2018). The limb sounder has a resolution in the horizontal domain of about 300 km along the line of sight and
30 km perpendicular to the line of sight (Fischer et al., 2008). Generally, the satellite data enable the measurement of the PSC
altitude and composition. Here, data on the occurrence of PSCs is obtained from both satellite instruments and sampled in the
vicinity of the measurement stations, i.e. for the Neumayer station data points within 68 °S to 73 °S and 0 °W to 20 °W are
considered and likewise for the Kiruna station data is taken from 66 °N to 71 °N and 10 °E to 30 °E.

## 3 Methodology

Polar stratospheric clouds lead to changes in the sky colour during twilight. This observation is used to identify PSCs using the
so-called colour index (CI) which is the ratio of measured radiances

$$\mathrm{CI} = \frac{I(\lambda_1)}{I(\lambda_2)} \tag{1}$$

at two wavelengths $\lambda_1 > \lambda_2$. The method takes advantage of the fact that scattering by particles has a characteristic wavelength
dependency (Sarkissian et al., 1991; von Savigny et al., 2005). Moreover, variations of the CI observed during twilight can be
attributed to particle scattering in the stratosphere as most tropospheric clouds are in darkness at solar zenith angles above 90°.
Nevertheless, thick tropospheric cloud layers still have an effect on the PSC signal.

It is important to choose the wavelengths such that these are not strongly influenced by Fraunhofer lines or molecular
absorption in the Earth's atmosphere. To avoid strong absorption by ozone, only wavelengths larger than 330 nm are considered.
In this study, the CI is calculated for the ratios of intensities at 390 nm to 340 nm in the UV as well as 610 nm to 444 nm in the
visible spectral range. Measured intensities are averaged in a ±1 nm bin around the respective wavelengths.

Previous studies used intensity ratios of 550 nm to 350 nm (Sarkissian et al., 1991) or 680 nm to 385 nm (Enell et al.,
1999, 2002). In those cases, the progression of the CI with SZA was analysed and a peak at about 93° to 94° was found if
PSCs were present above the instrument (Kiruna, Sweden). In Sarkissian et al. (1991) and Enell et al. (1999), the CI was
normalised to its value at SZA = 90° to compensate for possible tropospheric cloud effects. However, in a later study (Enell
et al., 2002) this was found not to be beneficial. The same finding could be confirmed in the current study.

A recent publication by Gomez-Martin et al. (2021) likewise uses the CI to infer PSC cases above the Belgrano II Antarctic
research station. They use the position of the CI maximum to retrieve information on the PSC layer altitude. However, high
temporal resolution of the measurements and special PSC conditions are needed to be able to infer the CI maximum with
sufficient accuracy (see Sect. 3.1).

For this study, the CI is averaged between 93.0° and 94.5° SZA and correlated to ECMWF temperature at 50 hPa pressure
level. The example of one year of data exhibits a distinct seasonal cycle (Sect. 4). To identify PSCs, a CI threshold is defined in





Sect. 3.2 based on the findings of the comparison of the CI with temperature and from the radiative transfer simulations which are discussed in the following.

## 3.1 Radiative Transfer Simulations using McArtim

Radiative transfer simulations are a key technique to evaluate the changes of the CI for various PSC scenarios. Using the
Monte Carlo radiative transfer model McArtim (Deutschmann et al., 2011, version of 21 April 2014), radiances are computed for the wavelengths 340 nm, 390 nm, 444 nm and 610 nm. The atmosphere in this model is represented by a pressure and temperature profile on a pre-defined altitude grid. The data is taken from the climatology by McLinden (2017) and considered at 75 °S. Also, ozone ($O_3$) volume mixing ratios are provided up to 100 km altitude. The absorption cross-section is taken from Serdyuchenko et al. (2014) and corresponds to a temperature of 223 K. Due to the pristine location of the stations, further trace
gases are neglected. Simulations are run for a detector at ground level with an opening angle of 0.3° and an elevation angle of 90° (zenith direction). The ground albedo is set to 0.8 (partially snow and/or ice covered). For each scenario, a series of simulations is calculated for different SZAs from 70° to 95° in steps of 0.5° (SZA > 90°) or 1° (SZA ⩽ 90°).

The scattering phase function $P(\theta)$ describing the angular scattering probability of a photon is only dependent on the scattering angle $\theta$ in case of spherical aerosol particles. It can be approximated by the Henyey-Greenstein function with an
asymmetry parameter $g = 0.7$ corresponding to particle sizes in the range of 0.3 $\mu$m to 10 $\mu$m (Laven, 2021). Further parameters in the simulations include the single scattering albedo ($\omega = 1$; Liou, 2002) and the Ångström exponent ($\alpha = 2$; Ångström, 1930).

McArtim is capable of 3D simulations. Nevertheless, due to the large horizontal extent of typical (non-lee wave) PSCs, the "standard" configuration chosen here computes a 1D scenario. This means that the simulation is performed on a 3D sphere but
the atmosphere as well as the PSC layer are horizontally homogeneous. The light path geometry is illustrated in Fig. 1a. While the station lies already in the shadow for SZA > 90°, photons may still traverse the PSC layer. Note that in the homogeneous case, the light always first traverses the PSC layer from above, whereas for the hemispheric PSC layer (Fig. 1b) or a local PSC (Fig. 1c) the photons can reach the PSC also directly from below before being scattered into the line of sight of the instrument. Here, the horizontal extent and altitude of the PSC layer are decisive. Therefore, simulations with different expansions beyond
the measurement site are considered. The impact of the geometric configuration and extent of the PSC layer is discussed at the end of this section.

Figure 2 depicts the simulation result of the "standard" configuration for the CI in the UV (390 nm to 340 nm; upper panel) and visible (610 nm to 444 nm; lower panel) spectral range. The PSC layer is simulated at a height $h_{PSC} = 20$ km to 22 km for different optical depths (at 1 $\mu$m) as given in the label. The scatter of the values as indicated by the $2\sigma$ range, i.e. two times the
standard deviation which should contain more than 95 % of all calculated spectra of a scenario, can be attributed to the noise introduced by the Monte Carlo method.

If no PSC layer is present ($\tau_{PSC} = 0$), an increase in CI is observed for SZA between 80° and 90°, which fits well with the every day experience of red sunsets/sunrises explained by the extinction due to Rayleigh scattering and ozone absorption in the atmosphere. At larger SZA, direct sunlight is scattered downward from high altitudes leading to a slight decrease in the CI.





In the presence of PSCs, different additional effects can be observed depending on the chosen spectral range of the CI. At SZA > 90°, the CI is reduced in the UV (upper panel) in the presence of a PSC. In the visible spectral range (lower panel), the effect depends on the optical thickness of the PSC layer. Here, a maximum is observed at approximately 93° to 95° for an optical depth of about $10^{-2}$. Low radiances at 610 nm lead to a reduced CI for optically thicker PSC layers.

Furthermore, the altitude of the PSC layer impacts the observed signal (Fig. 3). It can be seen that PSCs at higher altitudes

exhibit more prominent features. Layers at 12 km to 14 km altitude or below, however, would hardly be susceptible for real measurements.

**Influence of Tropospheric Clouds**

Moreover, tropospheric clouds influence the measured spectra of scattered sunlight. To qualitatively describe their effect, Fig. 4 compares a simulation with and without a tropospheric cloud. The tropospheric cloud layer is described by a 1 km thick layer

at 3 km to 4 km altitude with an optical depth of 5. Although the optical depth of tropospheric clouds can reach higher values, the impact is already clearly visible. The important parameter here is that the cloud is located directly above the measurement station and thus influences the light path between the PSC and the instrument. Cloud shadows with a gap above the instrument seem to have little influence (not shown). In both spectral ranges, the CI is enhanced in the case of an additional tropospheric cloud layer in the simulation. The effect is meanwhile larger at SZA ⩽ 90°. Nevertheless, this has to be considered when

looking at CI variations in measurement data also for SZA > 90°. On the one hand, PSCs might remain undetected in the UV when the enhancement due to tropospheric clouds is so large that the CI reduction in the presence of PSCs is completely compensated. On the other hand, tropospheric clouds affect the CI even in cases without PSC layer which in turn could affect the sensitivity to optically thin PSCs in the visible spectral range since they cannot be readily distinguished from the non-PSC case.

**Influence of the PSC Horizontal Extent**

The horizontal extent of the PSC is impacting the simulations, as pointed out by Laura Gomez-Martin and Daniel Toledo (personal communication, 2021). In order to investigate this effect, 3D simulations (Fig. 5) are performed according to the schematics in Fig. 1. Hereby, a PSC with a confined extension of 2° × 2° (compare to Fig. 1c) matches the result of the publication by Gomez-Martin et al. (2021). Interestingly, a PSC of this area leads to enhanced CI also in the UV which is

the opposite signal to what is found with a homogeneous PSC layer. In the visible, the CI maximum is further amplified. Additionally, the clear relationship between the height of the PSC layer and the position of the CI peak with respect to the SZA (see Fig. A2) allows to obtain information on the layer altitude for these PSC conditions. The 3D (1° hemisphere) simulation considers a PSC which extends from about 1° off the station towards the pole (compare to Fig. 1b). Note that the sun geometry is set such that the sunlight traverses the atmosphere from the equator. It yields similar results to the confined

PSC case. However, extending the hemispheric geometry to 5° off the station, the result shows some intermediate version between the confined PSC and homogeneous PSC cases. This can be explained by the fact that 5° corresponds to a dis-



tance between the station and the edge of the PSC layer of approximately $555\,\mathrm{km}$ $\left(= 2\pi R_E \cdot \frac{5°}{360°}\right)$, for which, at an SZA of $95°$, the photons originating from the sun are just able to "hit" the PSC layer from below. For an even larger PSC extent, e.g. $10°$, the simulations yield basically the same results as for the homogeneous PSC layer. According to Noel et al. 190 (2008) this is not unlikely for the Antarctic instrument where the PSC area can extend to about $65\,°\mathrm{S}$ during mid-winter.

Generally, the PSC layer altitude, optical depth and horizontal extent are the main drivers of CI variations for a given simulated wavelength ratio. A modified Ångström exponent is changing the optical depth, i.e. for $\alpha = 0$ the extinctions are slightly lower. However, the effect of changing the parameter settings remains small for both spectral ranges. For example, 195 the influence of the considered scattering phase function is investigated by running simulations with different asymmetry parameters $g$ of the Henyey-Greenstein phase function as well as a computed Mie phase function. The Mie phase function is calculated by a MATLAB routine (Mätzler, 2002) based on the formalism of Bohren and Huffman (1983) using a complex refractive index $m = 1.33931 + 7.29i \times 10^{-10}$, single log-normal particle size distribution $n(r)$ at $r = 1\,\mu\mathrm{m}$ with a standard deviation of $30\,\%$, and a total number of particles of $N = 10000$. Results are presented in the Appendix (Fig. A3 and A4). 200 Additionally, the effect of different atmospheric pressure and temperature profiles is examined. Other parameters changes, such as in the surface albedo, are not shown, but the effects are negligible.

To conclude, although the CI is not absolutely calibrated, systematics can be observed which allow the detection of PSCs from measured spectra. Thereby, the CI behaviour depends on the chosen wavelength ratio and the characteristics of the PSC layer like its altitude, optical depth and horizontal extent. Tropospheric clouds can influence the analysis, while other 205 parameters are of less importance. The simulation results for horizontally extended PSCs are compared to measurement data in the following to verify the applicability of the method for this type of PSC.

### 3.2 Comparison between Simulations and Measurements

Based on the knowledge obtained from the radiative transfer simulations, a CI threshold can be defined for the detection of PSCs from the measured spectra. In order to do so, for each year the minimum (in the UV) or maximum (in the visible) 210 CI averaged between $93.0°$ and $94.5°$ during the summer months is determined. For the Neumayer station the months from February until April are considered, whereas for the Kiruna station the value is retrieved from the months July to October. The threshold values are then averaged over the complete time series whilst taking into account instrumental changes. This means that an average threshold value is deduced for the period 1999 to 2002 and 2003 to 2021 for the Neumayer station as well as 1997 to 2006 and 2013 to 2021 for the Kiruna station. The error is given by the standard deviation of the annual thresholds. It 215 should be noted that for the Kiruna data two times the standard deviation is taken for the new detector system since 2013 due to the lower detector noise. Considering an averaged CI threshold allows for equal treatment of all years in the time series.

Data can then be categorised into PSC cases and non-PSC cases according to the measured CI between $93.0°$ and $94.5°$ SZA and the respective threshold value. The mean CI behaviour is compared to the simulation results in Fig. 6a for one year of data from the Neumayer station. Hereby, the 1D simulations with horizontally extended PSCs are chosen since these are expected 220 in the Antarctic vortex. It shows a good overall agreement. Particularly in the visible spectral range, results also match the





findings in Sarkissian et al. (1991) and Enell et al. (1999, 2002). The averaged PSC cases may thereby correspond to a PSC layer with an optical depth of 0.005 to 0.01 in the altitude range 20 km to 22 km. However, the observed signal could also originate from higher PSC layers with lower optical depths and vice versa since the simulations yield similar results for these cases. In any case, the exact determination of the PSC characteristics is not possible since the average value over the entire annual cycle is investigated.

Nevertheless, the comparison between simulations and measurements indicate a good applicability of this method to retrieve PSC cases from the measured spectra. A similarly good agreement is also found for the Kiruna station (Fig. 6b) giving confidence that the method is not only applicable for the persistent PSCs in the Antarctic but also the more variable conditions in the Arctic.

## 4   Discussion of the Seasonal Cycle

The correlation between CI and temperature is depicted in Fig. 7 (UV) and 8 (visible) for the Neumayer DOAS and in Fig. 9 for the Kiruna DOAS. The vertical dashed lines indicate the temperature thresholds for the formation of NAT (Type I) and ice (Type II) PSCs. The thresholds are taken from Larsen (2000) considering 10 ppbv $HNO_3$ (for NAT PSCs) and 5 ppmv $H_2O$. Variations of about 20 % in the mixing ratios of either gas correspond to $\pm 1$ K in the temperatures. The CI thresholds for the classification of PSC and non-PSC cases are indicated by a horizontal line.

A distinctive seasonal cycle is seen for either station. It is observed that PSCs form at altitudes above 20 km where temperatures first drop below their existence temperature. The cloud layers then shift downwards where temperatures remain cold even towards spring time. Thereby, continuous reformation of PSC particles is happening in the course of the winter. However, the radiative transfer simulations indicate only very low sensitivity of the CI to PSC layers at altitudes of about 12 km to 14 km or lower.

Moreover, it should be doubted whether the existence of PSCs as late as October or even November at the Neumayer station is reasonable considering the sedimentation of PSC particles (e.g., Tritscher et al., 2021). Not only does sedimentation move existing PSC layers to lower altitudes until the particles eventually evaporate, but the resulting denitrification and dehydration in the polar stratosphere prohibits reformation of new PSC layers despite low temperatures.

In the case of the Kiruna data set, additionally, many presumed PSC occurrences in spring (March/April) where temperatures above the station are high indicate another reasoning for the observed signal, namely the influence by volcanically induced aerosol particles. This is discussed in more detail in Sect. 6.1.

At the example of the Neumayer data, two further aspects can be studied. Firstly, how the presented approach compares between the UV and visible spectral range and secondly, how tropospheric cloud layers affect the retrieval. In the visible spectral range, the CI values suggest less PSCs during mid-winter. However, it is likely that optically thick PSCs remain undetected by this method as discussed in the previous section and thus lead to an underestimation of the PSC occurrence for that spectral range.





Because tropospheric clouds generally enhance the CI in both spectral regions, opposing effects are expected. The applied cloud filter algorithm is detailed in the Appendix A. In the UV, the filter has no effect on the CI threshold where the minimum

CI during the summer months determines the threshold value. However, there is a masking effect of tropospheric clouds to PSC cases. On the contrary, the influence in the visible is mainly on the derived CI threshold which is higher for the unfiltered data and thus reduces the sensitivity of this method to optically thin PSC layers. In both cases, cloud filtering leads to an underestimation of the detected PSC cases. Nonetheless, the course of the seasonal cycle is not affected by tropospheric cloud contamination.

In conclusion, the reduced (in the UV)/increased (in the visible) CI values during mid-winter can likely be attributed to PSC particles whereas for early spring possibly other "residual" particles lead to enhanced scattering at high altitudes since there is no indication for instrumentally induced artefacts. As pointed out above, the latter will be discussed in Sect. 6.1 exemplarily for the Kiruna data. While the seasonal cycle is discussed here at the example of an individual year, the main findings apply also more generally as can be seen from the investigation of the entire time series (see Fig. 11 and Fig. 12 for Neumayer and

Kiruna, respectively).

## 4.1 Comparison to Satellite Data

For a comparison to satellite data, the relative frequency of PSC signals in the measurements is calculated on a monthly basis for each DOAS instrument. Additionally, the number of PSC sightings in the vicinity of the measurement stations (68 °S to 73 °S and 0 °W to 20 °W for Neumayer as well as 66 °N to 71 °N and 10 °E to 30 °E for Kiruna) is retrieved from the CALIOP

and MIPAS PSC climatologies and the relative frequency is calculated for the respective months to obtain the same quantity as derived from the DOAS measurements. Most months have 15 or more days of satellite observations within the considered area around the ground-based stations, providing good data coverage for determining the PSC relative frequency. Due to the large measurement domains of MIPAS, however, only one or two data points are given per day, while CALIOP can provide more than 300 profiles daily.

The resulting comparison of one year of data for both stations is presented in Fig. 10a (Neumayer) and 10b (Kiruna). All data sets exhibit a distinct seasonal cycle.

The CALIOP data for the location of the Neumayer station shows highest PSC occurrence during July with nearly 100%, while MIPAS as well as the UV data suggest high occurrences even towards September and October. Generally, there is good agreement between all four data sets. Specifically, it should be noted that PSC sightings could deviate significantly within the

280 sampling area, although the sampling of the satellite data is restricted to a 300 km × 300 km area around the measurement station. Obviously, the visible spectral range detects less PSCs, especially from June to August. However, the overall tendency of the seasonal cycle is in good agreement to the data retrieved in the UV spectral range. As such, this figure summarises well the most important observations so far: The sensitivity of the visible spectral range to PSCs is lower than that of the UV spectral range for the approach used in this work. In general, the data (especially in the UV) agrees well with retrievals from

285 CALIOP and MIPAS. However, higher PSC occurrences are observed by the DOAS instrument towards spring time.





For the comparison at Kiruna, only CALIOP data is chosen since there is no temporal overlap between CALIOP, MIPAS and the DOAS data. Nonetheless, there is a good agreement between the CALIOP retrieval and the PSC occurrence from the DOAS instrument as seen from Fig. 10b.

The comparison looks similar also for other years. While the occurrence of PSCs is underestimated in the DOAS data during early winter, the UV data of both stations show particularly high PSC occurrences in early spring.

## 5 Time Series of the PSC Occurrence above Neumayer, Antarctica

Figure 11 shows the distribution of the PSC relative frequency on a biweekly basis for the complete measurement time series of the DOAS instrument at the Neumayer station. For each two-week period, at least ten valid measurements are available, which corresponds to about 36 % of the possible sample rate of two measurements per day. Most of the time, however, the data coverage is higher, which ensures good statistics in the frequency calculation. The colour contours indicate the temperatures which are important for the formation of PSCs: below 196.0 K at 50 hPa NAT (Type I) PSCs form in the polar stratosphere, for even colder temperatures (< 188.5 K at 50 hPa) ice particles (Type II PSCs) can exist. In the UV spectral range, a high PSC occurrence is found throughout the winter season with a seasonal cycle as discussed in Sect. 4. Unexpectedly elevated PSC frequencies are found for many years even in late October and beginning of November. In the visible spectral range, the fact that optically thick PSC layers result in reduced CI, and thus remain undetected, leads to lower PSC occurrences.

On average, for the complete time series, the relative frequency of PSC detections amounts to about 37% in the UV and 21% in the visible spectral range. The yearly averages calculated between May and November are given in Table 1. There is no significant trend throughout the measurement period. Also, the ECMWF temperature at 50 hPa indicates no temperature trend at the instrument location.

Several years feature specific atmospheric conditions which are discussed individually. In 2002, a stratospheric sudden warming led to a vortex split in September that year (Charlton et al., 2005; Frieß et al., 2005). The rapid increase in temperature and thus reduced PSC sightings are also reported by the DOAS data (see respective panel in Fig. 11). Similarly, in 2019 the sudden warming of the stratosphere (Lim et al., 2020) leads to a reduced PSC occurrence above the station.

Aerosol particles induced from volcanic plumes enhance the stratospheric background aerosol and can influence the retrieval, possibly leading to higher than usual levels in the PSC relative frequency in spring 2015 following the Calbuco eruption in southern Chile (Bègue et al., 2017). A more detailed discussion is given at the example of the Kiruna data set in Sect. 6.1.

While the error on the calculated CI can be assumed to be small (< 1 %), tropospheric cloud contamination is an important factor of uncertainty. To estimate the error, the analysis for the Neumayer station data is also done with an applied cloud filtering (see Appendix A). In Sect. 4 it is discussed that the cloud filter leads to an underestimation of the PSC occurrence. For the entire time series, the retrieved PSC relative frequency is about 20 % higher in the UV when applying the filter. In the visible, the influence is slightly smaller. Here, cloud-contaminated spectra increase the CI threshold which counterbalances the higher number of otherwise falsely identified PSC cases.





## 6 Time Series of the PSC Occurrence above Kiruna, Sweden

The time series of PSC occurrences above Kiruna is shown in Fig. 12. The data gaps and greater uncertainties make it difficult
to interpret the data for the years prior the detector exchange in 2013. Afterwards, yearly cycles with smaller errors can be
observed. However, the detected number of PSC cases is quite variable due to the more unstable Arctic vortex. It can be seen
from the shading that temperatures seldom reach below the formation thresholds for PSC particles above the station. It should be
noted that the vortex minimum temperature can deviate highly from the temperatures in the vicinity of the measurement station
and thus moving air masses could still transport PSCs into the area without maintaining the cold temperature. Furthermore,
wave-induced PSCs are an important contributor to total PSC occurrence in the Arctic.

On average, a PSC relative frequency of about 18 % is detected above Kiruna. This is about half the value observed for the
Antarctic measurements. Despite the very different measurement locations and meteorological conditions, also for Kiruna no
trend is seen in the PSC occurrence (compare Table 2).

High PSC-like signatures are also found towards spring time (e.g. 2014/15, 2018/19 or 2019/20). This feature appears even
more often than what is observed from the Neumayer data set. Hereby, a comparison to ECMWF or also MERRA2 (Gelaro
et al., 2017) data shows reduced total ozone columns above Kiruna during those spring seasons. It indicates that possibly
particles similar to those of PSCs provide a surface for heterogeneous reactions and in turn impact the ozone chemistry.

LIDAR measurements at the ALOMAR station in Norway were used to analyse stratospheric background aerosol and
characterise its seasonal cycle in a study by Langenbach et al. (2019). Although slightly enhanced values are found in April,
there is no indication of a temporal coincidence to the DOAS signal. For this, also the high aerosol values in August to October
should be detected in the presented DOAS retrieval but are not. Natural seasonal variation of the stratospheric background
aerosol can thus be excluded as possible explanation of the spring time signature.

Another, more fitting explanation for the observed signatures are volcanically induced aerosol particles. A discussion on the
coincidence of volcanic eruptions and the DOAS signal are presented in the next section.

### 6.1 Influence of Volcanic Aerosol

The observed PSC-like signatures above Kiruna are found especially in years with preceding volcanic eruptions in the tropics
or the northern hemisphere. To get an idea on how the volcanic plumes are transported into high northern latitudes, satellite data
is investigated. In Fig. 13 the aerosol extinction coefficient measured by OMPS-LP is shown (Malinina et al., 2021). OMPS is
on-board the Suomi National Polar-orbiting Partnership (SNPP) satellite launched in late 2011 by NASA and its limb profiler
(LP) is used to retrieve information on stratospheric aerosols (Seftor et al., 2014). It is a very sensitive sensor due to the long
light path, the measurement geometry and the usage of the forward scattering peak.

From the longitudinally averaged aerosol distributions, the volcanic plumes can be tracked and a rough estimate on their
effect on the northern high latitudes can be accomplished. Three eruptions in particular are likely observed by the DOAS
instrument. These are the Kelut eruption on 13 Feb 2014 (7.55 °S, 112 °E) together with the Sangeang Api eruption (30 May
2014; 8.2 °S, 119.07 °E) marked by the yellow arrow in Fig. 13, the Ambae eruptions on 6 Apr and 27 Jul 2018 (15.4 °S,





167.84 °E; blue arrow), and the Raikoke eruption on 22 Jun 2019 (48.3 °N, 153.4 °E; purple arrow). For all cases, the OMPS-LP data shows high extinction values in the 30° to 60° latitude-band and a high PSC-like signature is found during spring time the following year.

Another large eruption was Calbuco on 22 Apr 2015 (41.19 °S, 72.37 °W). It is the only large eruption in the southern mid-latitudes. However, it does not show up in the DOAS data from Kiruna which is not surprising given the long interhemispheric exchange times. Nonetheless, a very high PSC occurrence was detected in the southern hemispheric winter of the respective year above the Neumayer station as indicated in Sect. 5. Volcanic aerosol particles may have led to this unusual enhancement.

All eruptions except the Calbuco eruption took place in the tropics or northern hemisphere where a strong dispersion linked with the seasonal cycle favour the distribution of volcanic aerosol particles by the Brewer-Dobson circulation in the atmosphere 360 (Kloss et al., 2020). The visual inspection of the vortex position (via the potential vorticity at 475 K potential temperature from ECMWF) indicates that enhanced signals correspond to phases where the vortex lies above Kiruna which could be explained by the compression of air inside the vortex and thus increase in aerosol extinction. On the contrary, wildfires (e.g. Jul-Aug 2015, number 5 in Fig. 13) exhibit no signatures in the DOAS measurements due to the different particle size distribution compared to volcanic aerosol or PSC particles.

**7 Conclusions**

The CI method yields the possibility to retrieve information on PSCs on a statistical basis from ground-based spectroscopic measurements for data sets acquired in both hemispheres.

This study presents the PSC occurrence for more than 20 years of data measured by DOAS instruments at the Antarctic research station Neumayer as well as a measurement station in Kiruna, Sweden. The spectra of zenith scattered sun light 370 enable the detection of PSCs by investigating changes in the CI during twilight. Thereby, the CI behaviour is dependent not only on the selected wavelengths but also the PSC properties like the extent, altitude and optical density of the cloud layer. This is examined in detail with the help of radiative transfer simulations using the model McArtim. It shows that depending on the chosen wavelength ratio an enhancement or reduction of the CI at high SZA ($> 90°$) is found in the presence of PSCs. For the special conditions of a localised PSC, the CI can even be used to infer the PSC altitude (see Gomez-Martin et al., 375 2021). However, the more common appearance of horizontally extended PSC layers drastically reduces the signal dependency on the PSC altitude. Nonetheless, the presented approach allows for a consistent investigation of PSC occurrences above the measurement stations. The impact of optically thick tropospheric clouds thereby lead to an underestimation of the detected PSC frequency.

The data show no trend in the frequency of PSC sightings. However, PSCs are observed twice as frequent above Neumayer 380 (37 %) as above Kiruna (18 %). It reflects the different meteorological conditions of Antarctic and Arctic measurements. Generally, a reasonable agreement is found to the satellite retrievals of CALIOP and MIPAS with largest deviation in the beginning and end of the respective winter season.





The unexpectedly high PSC-like signatures during spring time which are observed mostly for the Kiruna data set could be explained by induced volcanic aerosol particles in the stratosphere after large eruptions in the tropics or northern hemisphere
of the preceding year. Other stratospheric aerosol with size distributions similar to PSC particles lead to similar signatures in the CI and thus influence the applied method. Nonetheless, the approach gives an indication on the PSC occurrence measured from ground-based spectroscopic instruments which yields a good complementary data set to other retrievals.

Moreover, the existence of particles in the polar stratosphere during spring time might have implications on the ozone chemistry and is not yet fully captured by the stratospheric aerosol climatologies from satellite data. While an extensive study
of the composition of those aerosol particles is not possible with the DOAS instruments, the easy to apply method enables the detection of PSCs not only for individual days but over entire measurement series, which is a decisive advance compared to measurements with limited temporal coverage like e.g. satellite observations.

*Data availability.* Measurement data are available upon request from the corresponding author. The auxiliary data are freely accessible online.

**Appendix A: Tropospheric Cloud Filter Algorithm**

Optically thick tropospheric cloud layers impact the measured radiances and thus the CI. Hereby, the cloud conditions can be spatially and temporally highly variable which makes it difficult to find suitable parameters for their identification. A cloud detection and classification scheme based on MAX-DOAS observations was developed by Wagner et al. (2014, 2016). It has the advantage that the influence of the atmospheric state can be directly attributed to each spectrum. Furthermore, the proposed
algorithm classifies various types of cloud scenarios. However, the approach cannot directly be applied to observations at high latitudes since the increased surface albedo affects the atmospheric radiative transport (see Wagner et al., 2016). Moreover, the existing cloud classification schemes are optimised for SZA $< 80°$, which is not appropriate for this study. Nevertheless, especially thick tropospheric clouds are of interest and therefore a modified cloud filtering scheme based on $O_4$ measurements is used.

The concentration of the oxygen collision complex $O_4$ is proportional to the square of the $O_2$ concentration. Therefore, the $O_4$ concentration can be well estimated because variations due to pressure and temperature fluctuations are small. Measurements of the $O_4$ absorption, e.g. expressed as slant column density (SCD), can then be used to identify changes in the atmospheric radiation transport, and thus cloud conditions, as proposed in Wagner et al. (2014). The observed $O_4$ absorption depends on the light path which in turn changes due to cloud properties and viewing geometry. Optically thick clouds will
generally lead to a large increase in the $O_4$ SCDs. Similarly, fog or snow at the surface might lead to extended light paths and likewise enhanced $O_4$ absorptions.

The $O_4$ SCDs are retrieved using the differential optical absorption spectroscopy (DOAS) principle (Platt and Stutz, 2008). It should be noted here that in the DOAS analysis a reference spectrum is considered to eliminate strong Fraunhofer lines in the





measured spectrum. Therefore, the analysis yields not an absolute value but the differential slant column density (DSCD) of

$O_4$. Nevertheless, within the period of time, where a fixed reference spectrum is considered, it is possible to retrieve information on cloud conditions (for details see below).

The spectral analysis of $O_4$ is accomplished for each time period with fixed reference using the DOASIS software (Kraus, 2003). The absorption bands of $O_4$ at 360 nm in the UV and 477 nm in the visible spectral range are used for the two spectrometers at the Neumayer station, respectively. Data are filtered for RMS $< 2 \times 10^{-3}$ to assure good fit results. Due to instrumental

effects or external influences (e.g. failure of the cooling system) eventually a new reference spectrum has to be chosen for separate time periods and the analysis is adapted accordingly. To harmonise the time series, the frequency distribution of $O_4$ DSCDs is computed for each analysis and the median is taken as offset between the analyses. Examples are given in Fig. A1 (a, c). Time periods with less than 60 data points are dropped from the data set, because a statistical analysis would not be reasonable. However, this affects only one winter month and about 12 % of the entire time series.

The resulting $O_4$ DSCDs are compared to the ECMWF cloud fraction data below 4 km altitude to exclude high-altitude cloud layers (or even PSCs). The correlation is shown in Fig. A1 (b, d). The ECMWF data are insufficient for identifying individual cloud-affected spectra as the spatio-temporal variability of tropospheric clouds can be rather high. Moreover, the geometric cloud cover is provided, whereas enhanced $O_4$ DSCDs are an indicator for optically thick clouds. Nevertheless, a statistical comparison enables the definition of a threshold on the $O_4$ DSCDs which is then used to filter out tropospheric cloud

events from the data.

In order to do so, the 75$^{th}$ percentile of $O_4$ DSCDs is calculated considering only data points where the ECMWF cloud fraction is smaller than 0.6. Data exceeding this threshold are considered to be contaminated by optically thick tropospheric clouds which reduces the number of valid data points by about 40 % to 60 %. As a consequence, high CI values are mostly filtered as expected.

The analysis with applied cloud filter is then used to estimate the error introduced by tropospheric cloud layers to the retrieval of PSC occurrences using the CI method (see Sect. 5).

For the Kiruna data set, no cloud filter is applied as no clear correlation between the $O_4$ DSCDs and ECMWF cloud fraction data is found. Because tropospheric clouds are less critical for the investigation of the CI threshold in the UV, no alternative filter is applied.

**Appendix B: Supplementary Figures**

This sections provides further results of the radiative transfer simulations.

*Author contributions.* TW, UF and BL developed the method. CFE and UR operated and maintained the instrument at Kiruna. CFE carried out initial analyses of the CI prior to this study. TW, UF and MG provided the measurement data and DOAS analyses. BL run the final data



analysis and performed the radiative transfer simulations with support by TW, SD and JP. BL prepared the manuscript with contributions
from all co-authors. The study was supervised by TW, UF and SD and discussions with all co-authors gave valuable feedback.

*Competing interests.*   Thomas Wagner is a member of the editorial board of Atmospheric Chemistry and Physics.

*Acknowledgements.*   Our sincerest thanks go to Rolf Weller and the countless scientists and technicians at the Neumayer station, who operated
and maintained the MAX-DOAS instrument over so many years.



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





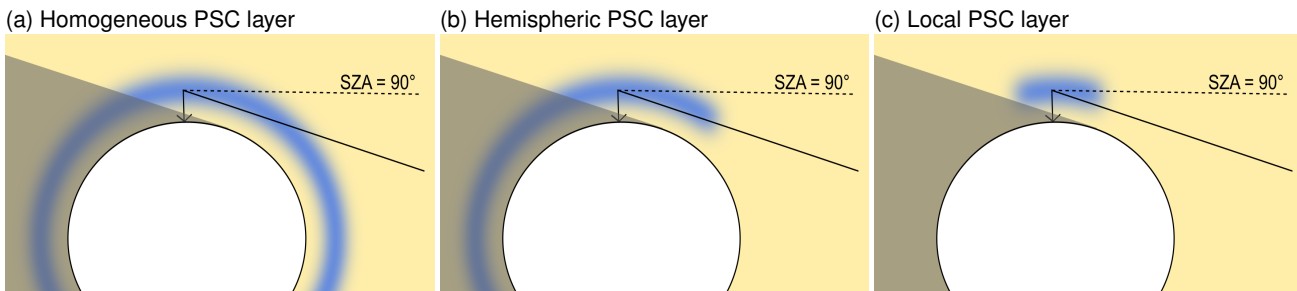

**Figure 1.** Scheme of different geometric configurations. The shadow indicates the area which cannot directly be "hit" by photons from the sun for SZA > 90°. For the PSC layer with hemispherical extent from the pole towards the station, the expansion beyond the measurement site is an important parameter. The effect is detailed at the end of Sect. 3.1.



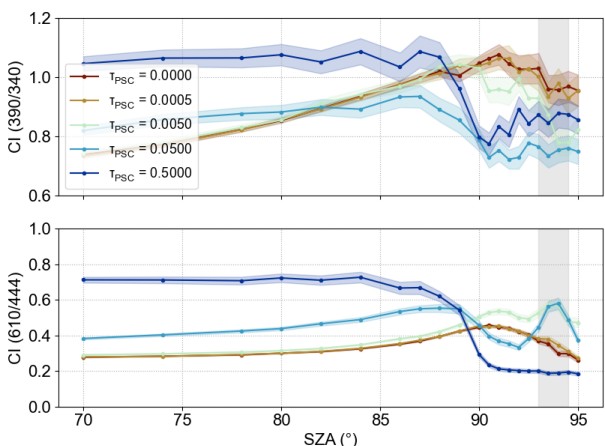

**Figure 2.** Simulation result for "standard" configuration (for details see text). Different colours correspond to different optical depths $\tau_{PSC}$ (at $1\,\mu$m) of the PSC layer ($h_{PSC} = 20$ to $22\,$km). The shading of the $2\sigma$ range indicates the noise of the results. The light grey shaded area depicts the SZA range ($93.0°$ to $94.5°$) which is used in the analysis of the measured spectra.





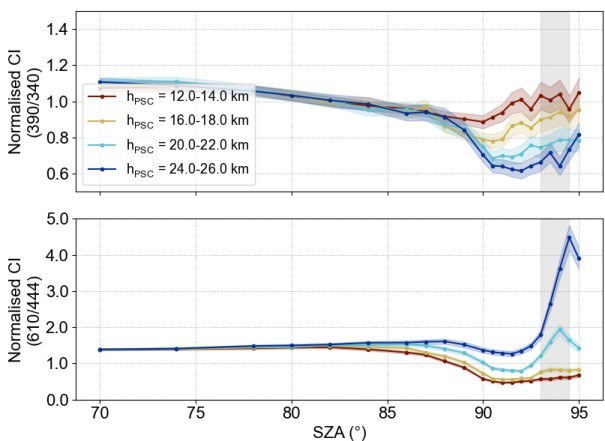

**Figure 3.** Simulation result (normalised CI) for different PSC layer heights $h_{PSC}$ ($\tau_{PSC} = 0.05$) for horizontally extended clouds. The colour index (CI) is normalised to the result without PSC layer for better comparison.



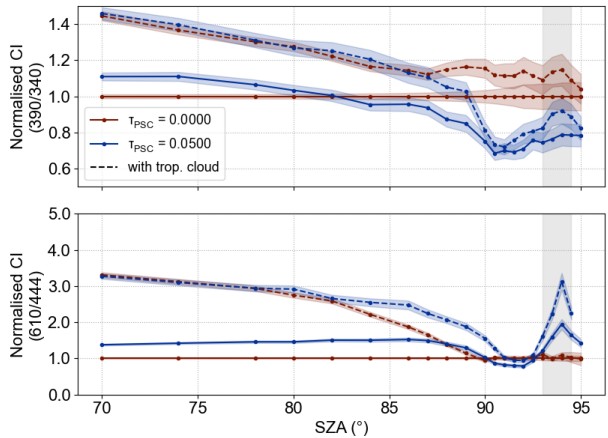

**Figure 4.** Simulation result (normalised CI) with a tropospheric cloud layer is shown for the non-PSC case ($\tau_{\mathrm{PSC}} = 0.00$) and a PSC layer ($\tau_{\mathrm{PSC}} = 0.05$) at $h_{\mathrm{PSC}} = 20$ to $22\,\mathrm{km}$. Dashed lines represent simulations with a tropospheric cloud layer at $3.0$ to $4.0\,\mathrm{km}$ (AOD = 5) compared to unaffected simulations (solid lines).



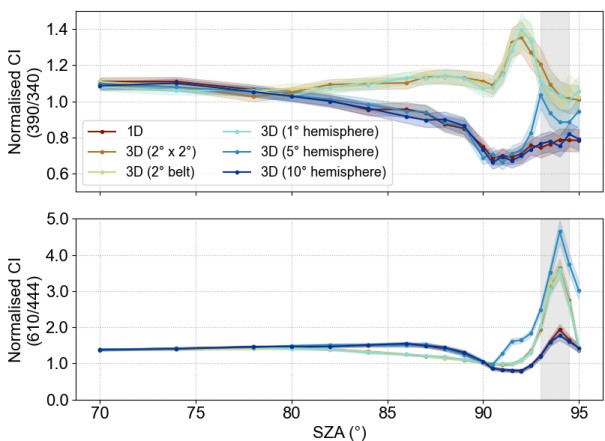

**Figure 5.** Simulation result (normalised CI) for different 3D geometries (compare to Fig.1). Details are given in the text. The PSC layer is described by $\tau_{PSC} = 0.05$ and $h_{PSC} = 20$ to $22\,km$.



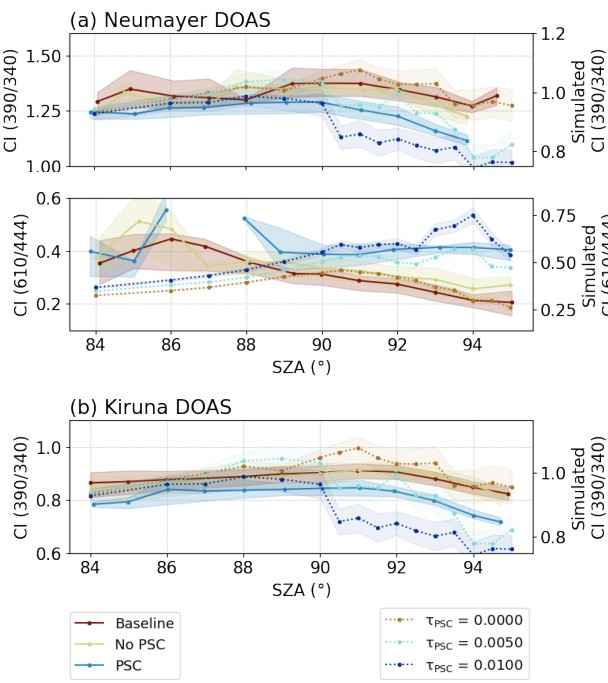

**Figure 6.** The average CI is depicted for the baseline (blue) as well as PSC (red) and non-PSC (orange) cases for the measurements at both stations. For the Neumayer station (a), the upper panel shows data in the UV and the lower panel in the visible spectral range. The baseline refers to data from February to April, while for the Kiruna station (b) it refers to data from July to October. Shading indicates the lower to upper quartile values of the data. Additionally, simulation results are depicted (dashed lines) for different optical depths $\tau_{PSC}$ at $h_{PSC} = 20$ to 22 km. Here, shading indicates the $2\sigma$ range of the results.



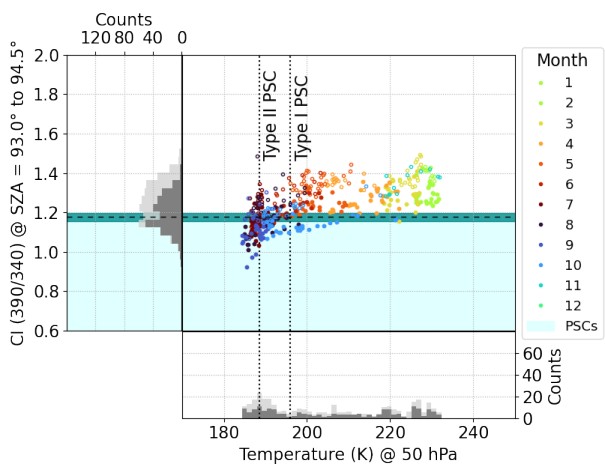

**Figure 7.** Colour index (CI) in the UV of the year 2007 for the Neumayer measurements. Data is plotted against ECMWF temperature at 50 hPa and colour-coded for the respective month of the measurement. The temperature thresholds for the formation of Type I and II PSCs are indicated by the dashed lines. The histograms depict the distribution of data points for the CI (left panel) and the temperature (bottom panel). Light grey distributions correspond to the unfiltered data, dark grey distributions represent the data filtered for tropospheric clouds, i.e. high $O_4$ DSCDs as detailed in the Appendix A. Likewise, the unfilled circles indicate data points that may be cloud-contaminated. The blue shaded area comprises the PSC cases. The error margin of the CI threshold is shaded in turquoise.



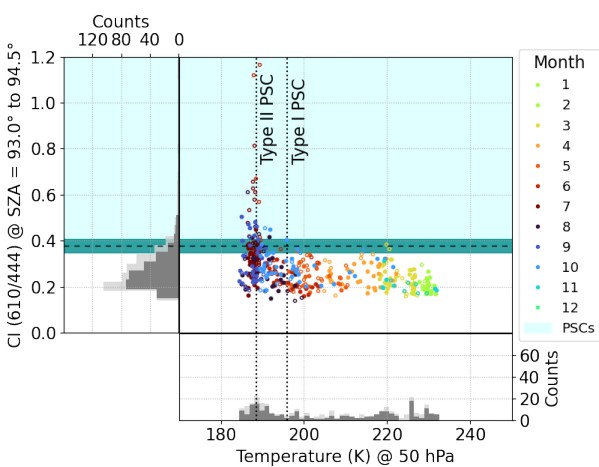

**Figure 8.** Colour index (CI) in the visible of the year 2007 for the Neumayer measurements. Data is represented as in Fig. 7. Note that PSC cases in the visible are characterised by an increased CI as opposed to a CI decrease in the UV.





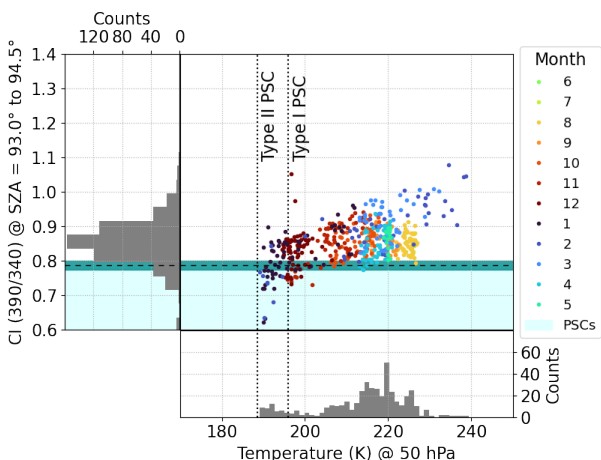

**Figure 9.** Colour index (CI) in the UV of the year 2017 for the Kiruna measurements. Data is represented as in Fig. 7, but no tropospheric cloud filter was applied.





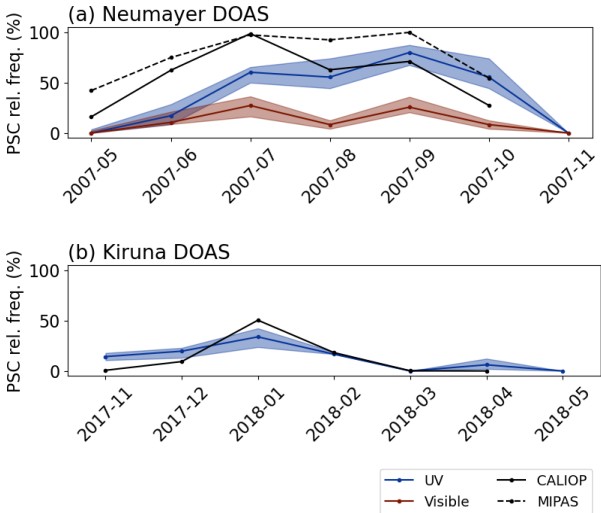

**Figure 10.** The occurrence of PSCs is depicted for the DOAS instruments (blue, red) as well as CALIOP (black solid) and MIPAS (black dotted) data above (a) the Neumayer station of the winter 2007 and (b) Kiruna of the winter 2017/2018.



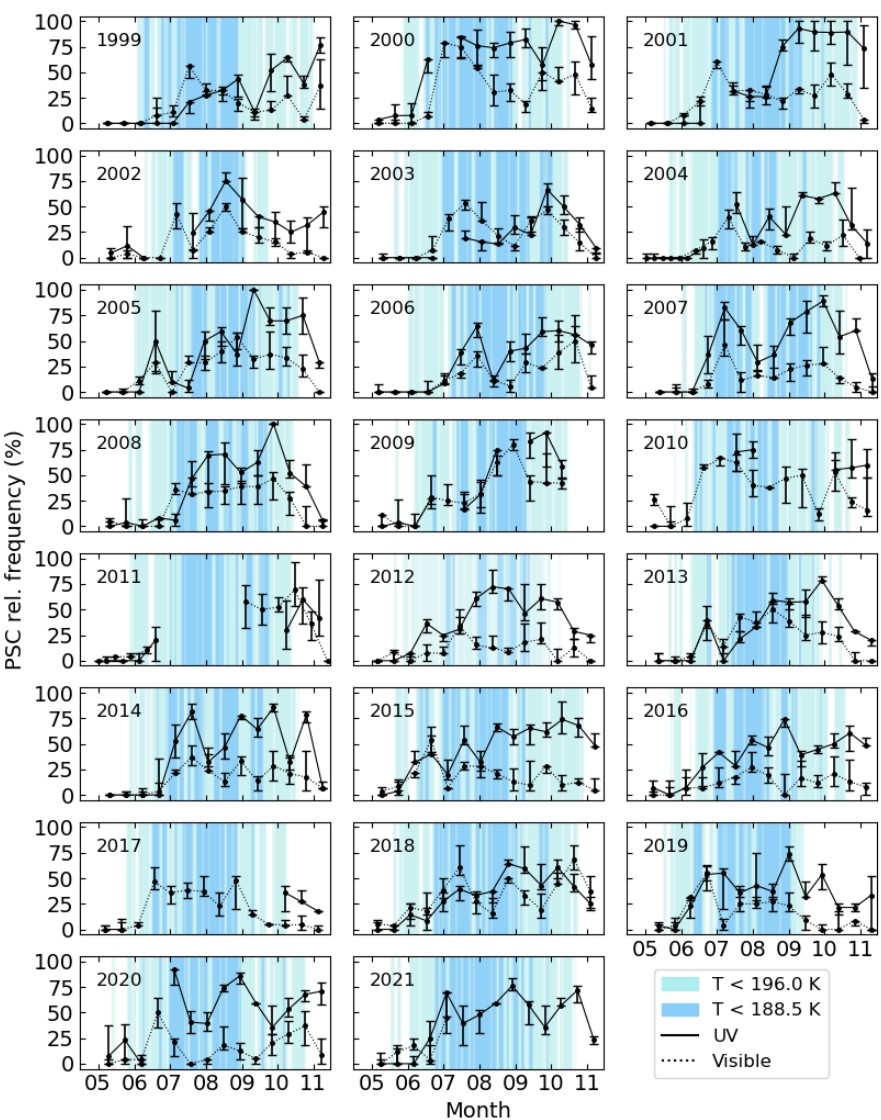

**Figure 11.** Biweekly relative frequency of PSCs in the UV (solid) and visible (dotted) spectral range for each year from May to November above the Neumayer station. In the background, the temperature ranges important for PSC formation are indicated by the blue shading. Data is plotted from the ECMWF subset (at 50 hPa) sampled in the vicinity of the measurement station.



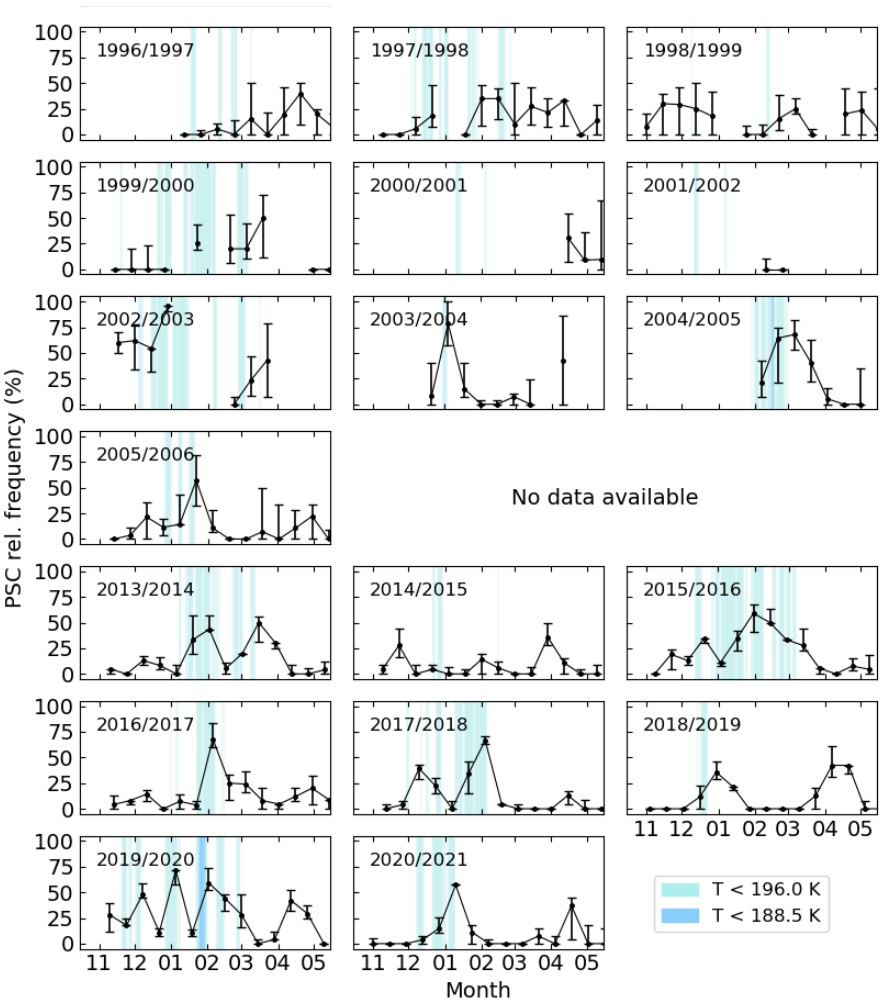

**Figure 12.** Biweekly relative frequency of PSCs in the UV spectral range for each year from November to May above Kiruna. In the background, the temperature ranges important for PSC formation are indicated by the blue shading. Data is plotted from the ECMWF subset (at 50 hPa) sampled in the vicinity of the measurement station.



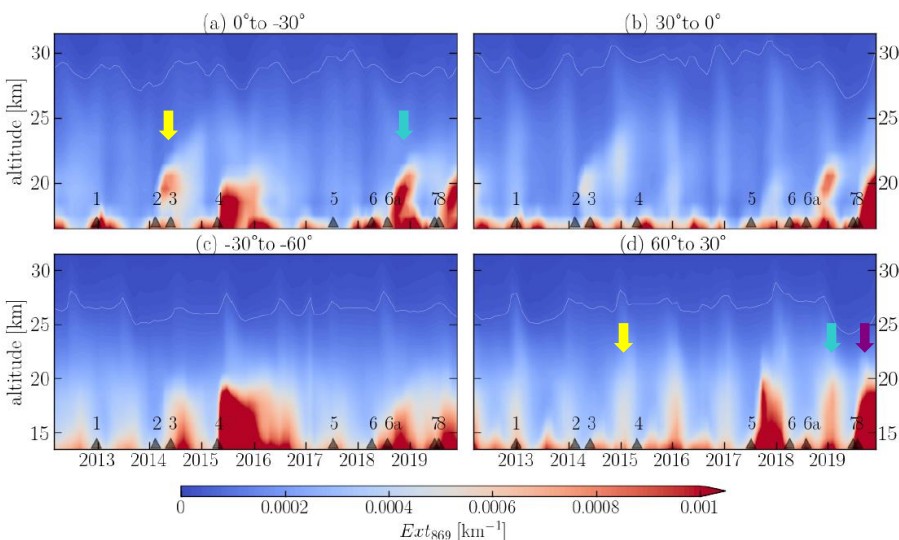

**Figure 13.** Monthly mean aerosol extinction coefficient ($Ext_{869}$) distribution from the OMPS-LP satellite instrument. The triangles represent volcanic eruptions and biomass burning events. Coloured arrows indicate the events which are important to this study. Adapted from Malinina et al. (2021).



**Table 1.** The average PSC relative frequency in % above the Neumayer station for each winter season from May to November. Data is given for both spectrometers as indicated.

| Year | Average PSC rel. frequency (%) | | Year | Average PSC rel. frequency (%) | |
|---|---|---|---|---|---|
| | UV | visible | | UV | visible |
| 1999 | $30\pm_6^4$ | $18\pm_4^7$ | 2011 | $24\pm_{12}^{14}$ | $29\pm_9^{10}$ |
| 2000 | $63\pm_{12}^9$ | $33\pm_6^6$ | 2012 | $38\pm_7^7$ | $11\pm_4^8$ |
| 2001 | $49\pm_{12}^{10}$ | $26\pm_5^4$ | 2013 | $33\pm_4^6$ | $21\pm_5^8$ |
| 2002 | $29\pm_{13}^{10}$ | $15\pm_3^3$ | 2014 | $43\pm_5^8$ | $16\pm_5^9$ |
| 2003 | $20\pm_4^5$ | $21\pm_4^6$ | 2015 | $45\pm_6^{10}$ | $18\pm_4^6$ |
| 2004 | $30\pm_4^{11}$ | $11\pm_4^4$ | 2016 | $38\pm_6^8$ | $12\pm_5^{12}$ |
| 2005 | $38\pm_8^8$ | $24\pm_6^7$ | 2017 | $16\pm_4^6$ | $20\pm_5^7$ |
| 2006 | $32\pm_5^6$ | $16\pm_5^6$ | 2018 | $35\pm_6^9$ | $33\pm_8^{10}$ |
| 2007 | $42\pm_{10}^9$ | $14\pm_4^7$ | 2019 | $34\pm_{10}^{10}$ | $16\pm_4^6$ |
| 2008 | $36\pm_8^{10}$ | $20\pm_7^7$ | 2020 | $49\pm_8^{10}$ | $16\pm_8^7$ |
| 2009 | $29\pm_9^9$ | $30\pm_7^{11}$ | 2021 | $40\pm_9^9$ | $15\pm_6^{11}$ |
| 2010 | $49\pm_7^{16}$ | $37\pm_8^9$ | | | |



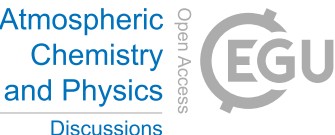

**Table 2.** The average PSC relative frequency in % above Kiruna for each winter season from November to May.

| Year | Average PSC rel. frequency (%) | Year | Average PSC rel. frequency (%) |
|---|---|---|---|
| 1996/1997 | $11\pm^{15}_{10}$ | | *[new detector system deployed]* |
| 1997/1998 | $17\pm^{12}_{12}$ | 2013/2014 | $15\pm^{7}_{5}$ |
| 1998/1999 | $16\pm^{18}_{14}$ | 2014/2015 | $8\pm^{6}_{4}$ |
| 1999/2000 | $13\pm^{13}_{6}$ | 2015/2016 | $22\pm^{6}_{5}$ |
| 2000/2001 | $10\pm^{26}_{7}$ | 2016/2017 | $15\pm^{6}_{7}$ |
| 2001/2002 | $16\pm^{14}_{3}$ | 2017/2018 | $15\pm^{4}_{4}$ |
| 2002/2003 | $53\pm^{11}_{18}$ | 2018/2019 | $12\pm^{4}_{4}$ |
| 2003/2004 | $15\pm^{19}_{8}$ | 2019/2020 | $26\pm^{7}_{6}$ |
| 2004/2005 | $35\pm^{16}_{16}$ | 2020/2021 | $10\pm^{6}_{5}$ |
| 2005/2006 | $12\pm^{15}_{8}$ | | |



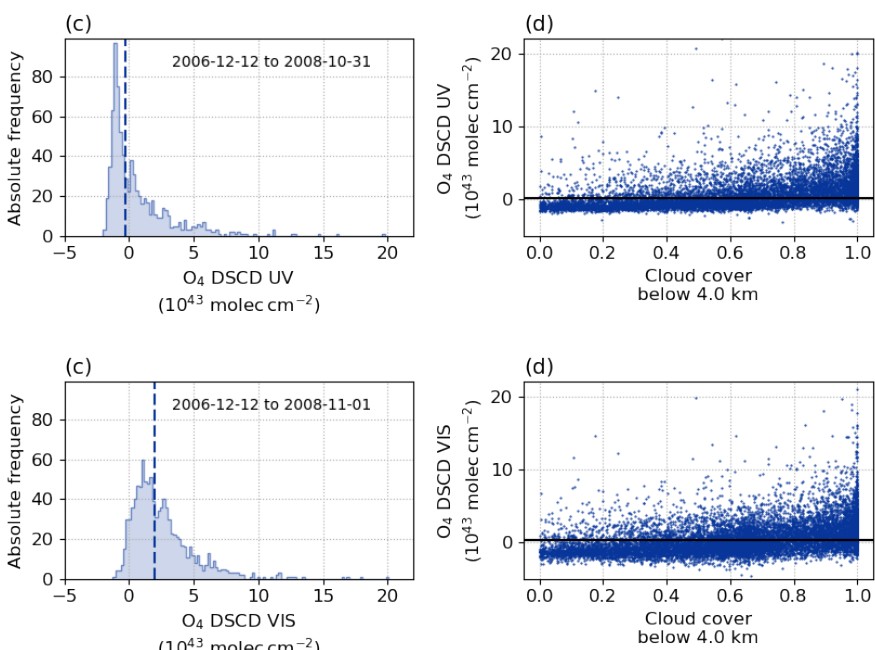

**Figure A1.** (a, c) Distribution of the O$_4$ DSCDs in the UV (upper) and visible (lower panel) spectral range for an analysis period in 2007. The median is indicated by a vertical line. (b, d) O$_4$ DSCDs in the UV (upper) and visible (lower panel) spectral range (1999-2018) are correlated to ECMWF cloud fraction data. The ECMWF data are averaged in the vicinity of the Neumayer station and the cloud cover below 4 km altitude is considered. The filter threshold (black line) is defined as the 75$^{th}$ percentile of the O$_4$ DSCDs considering only data points where the ECMWF cloud fraction is $< 0.6$.



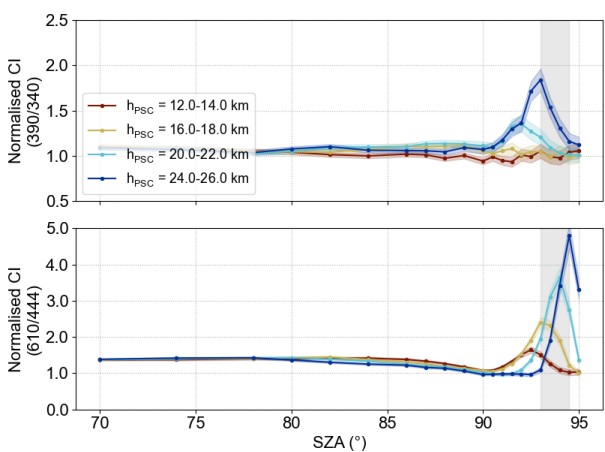

**Figure A2.** Simulation result (normalised CI) for different PSC layer heights $h_{PSC}$ ($\tau_{PSC} = 0.05$). The PSC is confined in a $2°$ x $2°$ area.





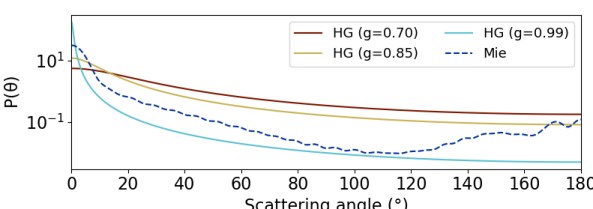

**Figure A3.** Comparison of different scattering phase functions $P(\theta)$. It shows different Henyey-Greenstein (HG) phase functions with their respective asymmetry parameter $g$ as well as a computed Mie phase function (details see Sect. 3.1).



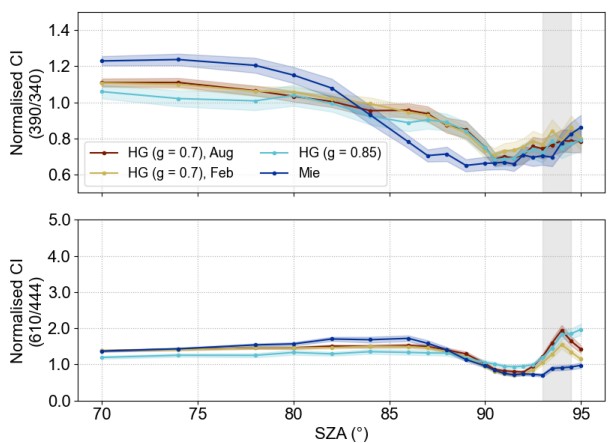

**Figure A4.** Simulation result (normalised CI) for different scattering phase functions and atmospheric profiles. The PSC layer is described by $\tau_{\mathrm{PSC}} = 0.05$ and $h_{\mathrm{PSC}} = 20$ to $22\,\mathrm{km}$. The considered phase functions are presented in Fig. A3.