# Peer review of "Occurrence of Polar Stratospheric Clouds as derived from Ground-based Zenith DOAS Observations using the Colour Index"

_Atmospheric Chemistry and Physics, 2022_

## Author Comment (AC1)

**Reply to comments from Referee #1 Alain Sarkissian**

We would like to thank Alain Sarkissian for the very positive and constructive comments which are addressed individually in the response below. The reviewer's comments are included in italics with the responses in blue.

**General Comments**

*This paper presents a very good application of ground-based zenith sky DOAS observation using the Colour Index. Scientific objectives are very well introduced as well as the instrumentation used, the methodology and modelling. Discussion of the influence of tropospheric clouds, by presence and by extend show cases that can be extended further.*
*The use of several wavelength ranges can help solving existing uncertainties and the radiative model simulations are very useful for it.*
*The two stations explored for this analysis, one in Antarctica the second in the Arctic are very well identified for PSC detection, and more, extended to volcanic aerosol detection, as discussed in this paper.*
*Conclusion reflects well the work done in this paper and the abstract also. The choice of putting appendices for the algorithm and for supplementary figures looks good for me.*

**Special comments**

*No comments for the language*

*Paragraph starting l-241: The authors have a discussion later in the text, please state it at the end of this paragraph*

In accordance to the remarks of the second referee, we have added a discussion point on the sensitivity of the PSC detection (l. 246 in the revised manuscript):
"Nonetheless, investigations of the sensitivity of the CI to different PSC properties as discussed in detail at the example of the simulation results indicate an under- rather than an overestimation of the detected PSC cases. It is, however, possible that particles of similar optical properties add to the PSC detection."
At the end of the section, the discussion is then only summarised.

*Figure 7: very good presentation*

Thank you!

*Figure 10: Remove DOAS from Neumayer and Kiruna titles of the figures because it is not only DOAS and put it in the UV DOAS and visible DOAS in legends*

Done.

*Figure 13: ...The triangles represent, I propose -> the black triangles at the bottom represent*

Done.

*Figure A1 and Appendix A could be in the main text, just before conclusion*

We have decided to keep the appendix since the discussion of the tropospheric cloud filter algorithm is important to the overall evaluation of the data quality and helps the interpretation, but it is not beneficial to the readability of the manuscript in this detail. Therefore, in the main part only the principal findings are stated with reference to the detailed description at the end of the manuscript for interested readers.

*Figures A2 to A4: I assume it should be Figures B1 to B3*

Done.

*Appendix B: please more text, the legends of the figure could be ok*

A description to the figures in the appendix was added (l. 448 in the revised manuscript):
"In Fig. B1 the height-dependency of the CI in case of a confined PSC (2° x 2°) is shown. The influence of different extents of a PSC layer is discussed in the respective subsection of Sect. 3.1. Figure B2 depicts different Henyey-Greenstein (with asymmetry parameter g) and the calculated Mie scattering phase functions. Details on the computation are given at the end of Sect. 3.1. The effect of changing the simulation parameters is depicted in Fig. B3."

---

## Author Comment (AC2)

**Reply to comments from Referee #2**

We would like to thank the referee for the very positive and constructive comments which are addressed individually in the response below. The reviewer's comments are included in italics with the responses in blue.

**General Comments**

*This work shows a systematic method to obtain information about the presence of PSCs using the colour index (CI) from the measured sky spectra. This method requires a previous survey at each location to obtain the threshold CI value averaged between SZA 93º and 94.5º that indicates the presence of PSCs. It also depends on the wavelength used to calculate the CI. A long series of data is required to obtain the appropriate statistics to apply this method.*
*The method is applied to experimental spectra recorded at two polar stations in the Arctic and Antarctic with a very impressive set of measurement data. The results are also compared with satellite data.*
*Applying this method, the seasonal occurrence of the PSCs for both stations is obtained and discussed and the distribution of the PSC occurrence for the whole data series is presented.*
*I find this work very consistent and recommend its publication with some comments. I find especially relevant the time series of PSC occurrence for such a long data series in these two different locations and the influence of volcanic aerosol in the PSC occurrence at Kiruna.*

**Special comments**

*Page 5, line 138. It is not clear to me what exactly the "standard" configuration is. Does it refer to a horizontally homogeneous PSC layer? In this case, the three scenarios defined in figure 1 are for a homogeneous PSC layer but with different extent? Later in the same paragraph, you mention again the homogeneous case referring to figure 1a. This is a bit confusing for me, it seems that homogeneous is not the same as the "standard" configuration. Please explain this in more detail in the text.*

The homogeneous PSC layer is chosen as standard configuration and the corresponding light path is shown in Fig. 1a. The other two illustrated scenarios are depicted to explain the differences in the geometric set-up and are discussed in more detail at the end of the RTM section. We have rephrased some sentences to make this clearer to the reader. The paragraph (starting at l. 138) now reads: "[…], the computation of a 1D scenario is chosen here as the "standard" configuration. This means that the simulation is performed on a 3D sphere but the atmosphere as well as the PSC layer are horizontally homogeneous. The corresponding light path geometry is illustrated in Fig. 1a. […] for other geometric configurations such as a hemispheric PSC layer (Fig. 1b) or a local PSC (Fig. 1c) the horizontal extent and altitude of the PSC layer are decisive. Therefore, additional simulations with different expansions beyond the measurement site are considered. The impact of the geometric configuration and extent of the PSC layer is discussed at the end of this section."

*Figure 1 caption. Please don't include an explanation about the interpretation of the figure in the caption, please, consider make this explanation in the text.*

We deleted the explanation from the figure caption and added a note on the standard configuration: "Scheme of different geometric configurations. The shadow indicates the area which cannot directly be "hit" by photons from the sun for SZA > 90°. The "standard" configuration refers to the homogeneous PSC layer (a). The effect of different other configurations is detailed at the end of Sect. 3.1."

*Figure 6. The caption doesn't correspond to the legend in the figure. In the caption blue means baseline as in the figure is red. Please, revise this.*

Thanks for pointing this out! We corrected the legend accordingly.

*On page 8 line 241, in the discussion on the occurrence of PSCs in November in Neumayer, have you considered that this method might overestimate the positive "detection" of PSCs? When compared to the satellite data below, the CI in the UV also overestimates the occurrence of PSCs in September with respect to CALIOP. Even the seasonal cycle captured by CALIOP and by this method are different. While CALIOP obtains a maximum occurrence in July, by this method the maximum is obtained in September. Then, looking at figure 11, I feel that the difference may be caused by the type of PSC. Perhaps the altitude of the layer is different for the different PSC types and your method is not capturing it (just lucubrating)?*

There are two important points which you mention here:
First, there remains an uncertainty in the detection of PSCs using the CI method. Since there is no absolute CI value for distinguishing PSC and non-PSC cases, the method relies on the qualitative comparison to simulation results and reasonable consideration of the seasonal cycle seen in the data. Investigations of different effects such as the masking by tropospheric cloud layers or the sensitivity of the CI to changes in the conditions (PSC altitude, optical depth, extent, etc. – compare to the simulation results), however, will cause an under- rather than an overestimation of the detected PSC cases.
Nevertheless, there is a discrepancy between CALIOP and DOAS measurements, with higher PSC relative frequency detected by CALIOP in the course of winter but lower frequency detected during spring time compared to the UV DOAS. Thus, secondly, the differences in the measurement techniques need to be considered. CALIOP is less sensitive to optically thin particle layers which could possibly explain the deviation during spring time. Further, the DOAS system is very sensitive to particles which fit the characteristic of typical PSC particles, also e.g. volcanic aerosol as discussed at the example of the Kiruna data set. Since the PSC type cannot be classified with the DOAS data, a more sophisticated comparison is unfortunately not possible at this point. In conclusion, uncertainties in both, the optical depth of the PSC layer and the particle type, make it difficult to determine the cause of the discrepancy.
Seeing the differences also between CALIOP and MIPAS, we would rate the overall agreement between ground-based and satellite-based measurements as reasonable while obviously not perfect.
We have added the points mentioned above to the manuscript:
L. 246 (in the revised manuscript): "Nonetheless, investigations of the sensitivity of the CI to different PSC properties as discussed in detail at the example of the simulation results will cause an under- rather than an overestimation of the detected PSC cases. It is, however, possible that particles of similar optical properties add to the PSC detection."
L. 285 (in the revised manuscript): "Moreover, because of the differences in the measurement platforms and data retrievals, perfect agreement is not to be expected."

*In page 9, line 280. You say that the seasonal cycle captured by UV-CI is in good agreement with the one captured by CALIOP, but as I commented previously, is not. UV-CI presents an absolute maximum in September whereas CALIOP records it in July in Neumayer and it is the same for Kiruna (but in January and April). Please, rewrite this.*

We have rephrased parts of this paragraph to more conservative statements, e.g. using "reasonable agreement" instead of "good agreement" (l. 283 in the revised manuscript) and added "with an absolute maximum in detected PSC cases in January" (regarding the PSC detection at Kiruna, l. 294 in the revised manuscript). Considering the different measurement platforms and data retrievals, the agreement seems reasonable to us (see also previous comment).

*Figures from A2 to A4, should be from B2 to B4.*

Done.